# Task-Agnostic Online Reinforcement Learning with an Infinite Mixture of Gaussian Processes

**Mengdi Xu, Wenhao Ding, Jiacheng Zhu, Zuxin Liu, Baiming Chen, Ding Zhao**

Carnegie Mellon University
{mengdixu, wenhaod, jzhu4, zuxinl, baimingc, dingzhao}@andrew.cmu.edu

## Abstract

Continuously learning to solve unseen tasks with limited experience has been extensively pursued in meta-learning and continual learning, but with restricted assumptions such as accessible task distributions, independently and identically distributed tasks, and clear task delineations. However, real-world physical tasks frequently violate these assumptions, resulting in performance degradation. This paper proposes a continual online model-based reinforcement learning approach that does not require pre-training to solve task-agnostic problems with unknown task boundaries. We maintain a mixture of experts to handle nonstationarity, and represent each different type of dynamics with a Gaussian Process to efficiently leverage collected data and expressively model uncertainty. We propose a transition prior to account for the temporal dependencies in streaming data and update the mixture online via sequential variational inference. Our approach reliably handles the task distribution shift by generating new models for never-before-seen dynamics and reusing old models for previously seen dynamics. In experiments, our approach outperforms alternative methods in non-stationary tasks, including classic control with changing dynamics and decision making in different driving scenarios. Codes available at: `https://github.com/mxu34/mbrl-gpmm`.

## 1  Introduction

Humans can quickly learn new tasks from just a handful of examples by preserving rich representations of experience [1]. Intelligent agents deployed in the real world require the same continual and quick learning ability to safely handle unknown tasks, such as navigation in new terrains and planning in dynamic traffic scenarios. Such desiderata have been previously explored in meta-learning and continual learning. Meta-learning [2, 3] achieves quick adaptation and good generalization with learned inductive bias. It assumes that the tasks for training and testing are independently sampled from the same accessible distribution. Continual learning [4, 5] aims to solve a sequence of tasks with clear task delineations while avoiding catastrophic forgetting. Both communities favor Deep neural networks (DNNs) due to their strong function approximation capability but at the expense of data efficiency. These two communities are complementary, and their integration is explored in [6].

However, real-world physical tasks frequently violate essential assumptions of the methods as mentioned above. One example is the autonomous agent navigation problem requiring interactions with surrounding agents. The autonomous agent sequentially encounters other agents that have substantially different behaviors (e.g., aggressive and conservative ones). In this case, the mutual knowledge transfer in meta-learning algorithms may degrade the generalization performance [7]. The task distribution modeling these interactions is prohibitively complex to determine, which casts difficulties on the meta-training process with DNNs [8, 9]. Additionally, the boundaries of tasks required in most continual learning algorithms cannot feasibly be determined beforehand in an online

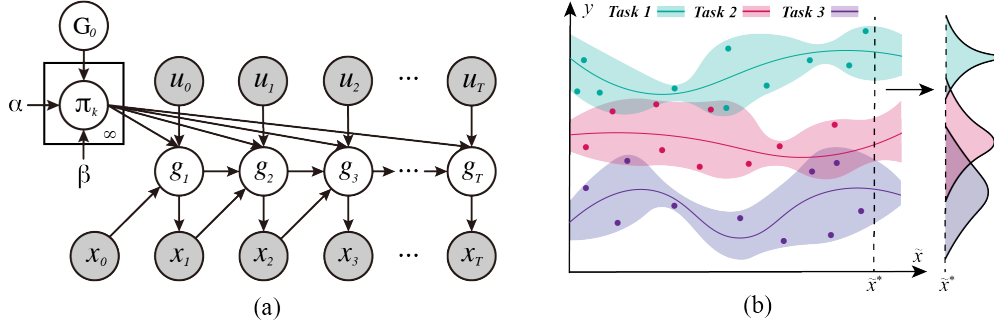

Figure 1: Method illustration. **(a)** is a graphical representation of the proposed model-based RL with an infinite mixture as the dynamics model. $u_t$, $x_t$, and $g_t$ represent the action, state, and the dynamics model at time $t$, respectively. Parameters include the concentration parameter $\alpha$, the base distribution $G_0$, and the sticky parameter $\beta$. **(b)** visualizes the predictive distribution at a data point $\tilde{x}^*$.

learning setting. Although task-agnostic/task-free continual learning is explored in [10, 11], the temporal dependencies of dynamics presented in a non-stationary robotics task are missed. For instance, two different dynamics models close together in time are likely to be related.

In this work, we aim to solve nonstationary online problems where the task boundaries and the number of tasks are unknown by proposing a model-based reinforcement learning (RL) method that does not require a pre-trained model. Model-based methods [12] are more data-efficient than model-free ones, and their performance heavily depends on the accuracy of the learned dynamics models. Similar to expansion-based continual learning methods [6, 13], we use an infinite mixture to model system dynamics, a graphical illustration of which is given in Figure 1 (a). It has the capacity to model an infinite number of dynamics, while the actual number is derived from the data. We represent each different type of dynamics with a Gaussian Process (GP) [14] to efficiently leverage collected data and expressively model uncertainty. A GP is more data-efficient than a DNN (as its predictive distribution is explicitly conditioned on the collected data) and thus enables fast adaptation to new tasks even without the use of a previously trained model. With a mixture of GPs, the predictive distribution at a data point is multimodal, as shown in Figure 1 (b), with each mode representing a type of dynamics. By making predictions conditioned on the dynamics assignments, our method robustly handles dynamics that are dramatically different.

At each time step, our method either creates a new model for previously unseen dynamics or recalls an old model for encountered dynamics. After task recognition, the corresponding dynamics model parameters are updated via conjugate gradient [15]. Considering that RL agents collect experience in a streaming manner, we learn the mixture with sequential variational inference [16] that is suitable for the online setting. To account for the temporal dependencies of dynamics, we propose a transition prior that stems from the Dirichlet Process (DP) prior to improve task shift detection. We select representative data points for each type of dynamics by optimizing a variational objective widely used in the Sparse GP literature [17]. We demonstrate the capability of task recognition and quick task adaptation of our approach in non-stationary `Cartpole-SwingUp`, `HalfCheetah` and `Highway-Intersection` environments.

## 2 Related Work

Meta-learning algorithms in general consist of a base (quick) learner and a meta (slow) learner [18, 19] and have recently been combined with RL to solve nonstationary problems via model-free [20, 21, 22] and model-based approaches [3, 2]. The closest work to our research is [8], which uses a mixture of DNNs as the dynamics model in model-based RL but still requires a model-agnostic meta-learning (MAML) prior. Researchers in [9] augment MAML with a multimodal task distribution for solving substantially different tasks. However, it inherits the limitations of MAML (such as assuming accessible task simulations and clear task boundaries). Additionally, the meta training strategy increases sample complexity and thus makes many model-free meta-learning algorithms infeasible to implement on real-world problems [23]. Using a single GP as the dynamics model in model-based meta-RL is explored in [2] by updating latent variables for different tasks but in an offline and

episodic setting. To the best of our knowledge, our method is the first capable of robustly handling online nonstationary tasks without requiring task delineations or depending on a pre-trained model.

Using a mixture of experts to solve different tasks is explored in continual learning [24, 4, 25] and Multi-task Learning [26, 27, 28]. However, previous works from both communities require clear task delineations. The conventional inference methods for mixture models mainly consist of MCMC sampling [29, 30] and variational inference [31]. Both methods keep the information of the whole dataset and do inference iteratively via multiple passes. In contrast, streaming variational inference [32, 33, 34] for Bayesian nonparametric models is designed for handling streaming data and requires a single computation pass. Sequential variational inference [16] for DP mixtures has been recently integrated with DNNs for image classification [11].

## 3   Model-Based RL with an Infinite Mixture of Gaussian Processes

Model-based RL algorithms rely on the learned dynamics model to roll out environments. For real-world tasks that contain substantially different dynamics, using an infinite mixture model as the dynamics model alleviates the harmful mutual knowledge transfer when using a single model, and enables the backward transfer of knowledge by recalling and updating stored dynamics models. Learning the system dynamics model $f$ is carried out by performing inference on data-efficient GPs to avoid training a prior model as in [6, 8]. Additionally, GPs define distributions over functions and thus naturally capture the aleatoric uncertainty of noisy environments. We use Model Predictive Control (MPC) for selecting an action at each time step, which can be seen as a closed-loop controller and increases robustness to model errors.

We consider the system dynamics model in the from of $\boldsymbol{x}_{t+1} = \boldsymbol{x}_t + f(\boldsymbol{x}_t, \boldsymbol{u}_t)$. The input is augmented as $\tilde{\boldsymbol{x}}_t = (\boldsymbol{x}_t, \boldsymbol{u}_t)$, where $\boldsymbol{x}_t \in \mathbb{R}^c$ and $\boldsymbol{u}_t \in \mathbb{R}^d$ are the state and action at time $t$, respectively. The target $\boldsymbol{y}_t = \Delta \boldsymbol{x}_t = \boldsymbol{x}_{t+1} - \boldsymbol{x}_t$ is the state increment. A history dataset $\mathcal{D} = \{\tilde{\boldsymbol{x}}^j, \boldsymbol{y}^j\}_{j=1}^m$ is maintained to update the mixture model and predict increments in MPC. The inputs and targets are aggregated into $\tilde{X} \in \mathbb{R}^{(c+d) \times m}$ and $Y \in \mathbb{R}^{c \times m}$. With each dynamics model as a GP [14], the dynamics function $f$ can be decoupled by dimension as $f = (f_1, ..., f_c)$ with $f_i : \mathbb{R}^c \to \mathbb{R}$. The state difference given a new observation $\boldsymbol{x}$ and action $\boldsymbol{u}$ is drawn from the predictive distribution

$$p(f|\mathcal{D}, \tilde{\boldsymbol{x}}_t) = \prod_{i=1}^c \mathcal{N}(\mathrm{m}(f_i), \mathrm{cov}(f_i)). \tag{1}$$

The mean function is $\mathrm{m}(f_i) = K_i(\tilde{\boldsymbol{x}}, \tilde{X})[K_i(\tilde{X}, \tilde{X}) + \sigma_i^2 I]^{-1} Y^i$ and the covariance matrix is $\mathrm{cov}(f_i) = K_i(\tilde{\boldsymbol{x}}, \tilde{\boldsymbol{x}}) - K_i(\tilde{\boldsymbol{x}}, \tilde{X})[K_i(\tilde{X}, \tilde{X}) + \sigma_i^2 I]^{-1} K_i(\tilde{X}, \tilde{\boldsymbol{x}})$. $Y^i$ denotes the target's $i$th dimension. $\sigma_i$ is the standard deviation of observation noise of dimension $i$. The matrix $K_i$ is fully specified by the kernel function $k_i(\tilde{\boldsymbol{x}}, \tilde{\boldsymbol{x}}_*)$, which defines the function smoothness. For simplicity, we use the scaled squared exponential kernel $k_i(\tilde{\boldsymbol{x}}, \tilde{\boldsymbol{x}}_*) = w_i^2 \exp(-\frac{1}{2} \sum_{j=1}^{c+d} w_{i,j}(\tilde{\boldsymbol{x}}^j - \tilde{\boldsymbol{x}}_*^j)^2)$. $w_{i,j}$ is the reciprocal of the lengthscale between dimensions $i$ and $j$, and $w_i$ is the output scale of dimension $i$.

## 4   Scalable Online Bayesian Inference

This section presents a scalable online Bayesian inference method for learning an infinite mixture of GPs in the model-based RL setting. We assume that it is possible for a new type of dynamics to emerge at every timestep, and the total number of different dynamics is unknown. We observe a new data pair $(\boldsymbol{x}_t, \boldsymbol{u}_t, \Delta \boldsymbol{x}_t)$ each time $t$ the RL agent interacts with the environment. To learn the mixture model, we first identify the latent dynamics assignment variable $z_t$ and then update the dynamics-specific parameters $\boldsymbol{\theta}_{z_t}$ of expert $M_{z_t}$. Our goal is to jointly do inference over $z$ and learn $\boldsymbol{\theta}$ in an online streaming setting by maximizing the log posterior probability given the observed data $p_n(z_{1:n}, \boldsymbol{\theta}|\mathcal{D})$. To get a reasonable action at time $t + 1$, with the inferred dynamics assignment $z_t$, we select the expert $M_{z_t}$ to generate predictions $\boldsymbol{y}_{t+1:t+T}$ for the MPC.

We first introduce sequential variational inference with transition prior in Section 4.1. To make our model scalable to large datasets, we introduce an optimization-based method in Section 4.2 to eliminate redundant data points. The stability of the mixture model is enhanced by merging similar

**Algorithm 1:** Bayesian Inference for Continual Online Model-based Reinforcement Learning

---

**Input:** Concentration parameter $\alpha$, Initial parameter $\boldsymbol{\theta}_0$, Sticky parameter $\beta$, KL threshold $\epsilon$, Merge trigger $n_{merge}$, Data Distillation trigger $n_{distill}$, Inducing point number $m$
**Output:** Infinite Mixture Model $\boldsymbol{M}$, Representative Dataset $\mathcal{D}$
Initialization: $\boldsymbol{M} \leftarrow \{M_0\}, \mathcal{D} \leftarrow \{\emptyset\}, t \leftarrow 0, z_0 \leftarrow 0$, and $K \leftarrow 1$;
$(\boldsymbol{x}_0, \boldsymbol{u}_0, \Delta\boldsymbol{x}_0) \leftarrow RandomPolicy$;
$\mathcal{D} \leftarrow \mathcal{D} \cup \{(z_0, \boldsymbol{x}_0, \boldsymbol{u}_0, \Delta\boldsymbol{x}_0)\}$ ;
Update $\boldsymbol{\theta}_0$ ;
**while** *task not finish* **do**

    $z_{old} \leftarrow z_t, t \leftarrow t + 1$;
    $(\boldsymbol{x}_t, \boldsymbol{u}_t, \Delta\boldsymbol{x}_t) \leftarrow MPC(M_{z_{t-1}})$;
    // Sequential Variational Inference with Transition Prior
        (Section 4.1)
    Update $q_t^{pr}(z_t)$ ;
    Update $z_t = \arg\max_k \rho_t(z_{tk})$ ;
    **if** $z_t = K$ **then**
        Append $M_{z_t}$ to $\boldsymbol{M}$, $K \leftarrow K + 1$
    $\mathcal{D} \leftarrow \mathcal{D} \cup \{(z_t, \boldsymbol{x}_t, \boldsymbol{u}_t, \Delta\boldsymbol{x}_t)\}$;
    Update $\boldsymbol{\theta}_{z_t}$ ;
    // Expert Merge and Prune (Section 4.3)
    **if** $\sum_{i=0}^{t} \mathbf{1}\{z_i = z_t\} = n_{merge}$ **then**
        $d_t(k) = d(M_{z_t}, M_k), \ k = 0, ..., K - 1$ ;
        **if** $\min_k d_t \leq \epsilon$ **then**
            Merge $M_{z_t}$ to the most similar model $M_{\arg\min_k d_t}$, $K \leftarrow K - 1$
    **if** $\sum_{i=0}^{t} \mathbf{1}\{z_i = z_{old}\} \leq n_{merge}$ **and** $z_t \neq z_{old}$ **then**
        Merge $M_{z_t}$ to the most similar adjacent model based on $d(M_{z_t}, M_k)$, $K \leftarrow K - 1$
    // Data Distillation (Section 4.2)
    **if** $\sum_{i=0}^{t} \mathbf{1}\{z_i = k\} \geq n_{distill}, \ \forall k$ **then**
        Get $m$ inducing points

---

experts and pruning redundant ones, as in Section 4.3. We present the overall pipeline in Algorithm 1 and discuss the effect of model parameters in Section S2 in the supplementary material.

## 4.1 Sequential Variational Inference with Transition Prior

We approximate the intractable $p_n(z_{0:n}, \boldsymbol{\theta}|\mathcal{D})$ as $\hat{q}_n(z_{0:n}, \boldsymbol{\theta}) = \prod_{k=0}^{\infty} \gamma_n(\boldsymbol{\theta}_k) \prod_{i=0}^{n} \rho_n(z_i)$ using Assumed Density Filtering (ADF) and mean field approximation. As derived in [34], the optimal distribution of the dynamics assignment $z_n$ of the $n$th observation is

$$\rho_n(z_{nk}) \propto \begin{cases} q^{pr}(z_{nk}) \int p((\tilde{\boldsymbol{x}}_n, \boldsymbol{y}_n)|\boldsymbol{\theta}_k)\gamma_{n-1}(\boldsymbol{\theta}_k)d\boldsymbol{\theta}_k & 0 \leq k \leq K_{n-1} - 1 \\ q^{pr}(z_{nk}) \int p((\tilde{\boldsymbol{x}}_n, \boldsymbol{y}_n)|\boldsymbol{\theta}_k)G_0(\boldsymbol{\theta}_k)d\boldsymbol{\theta}_k & k = K_{n-1} \end{cases} \qquad (2)$$

where $q^{pr}(z_n) = \sum_{z_{0:n-1}} p(z_n|z_{0:n-1}) \prod_{i=0}^{n-1} \rho_{n-1}(z_i)$, and $G_0$ is the base distribution for dynamics-specific parameter $\boldsymbol{\theta}$. $q^{pr}$ acts as the prior probability of the assignments and is analogous to the predictive rule $p(z_n|z_{0:n-1})$. $G_0$ and $\gamma_{n-1}$ in general are in the same exponential family as $\rho_n(z_n)$ so that (2) is in closed form. When using a DP mixture model, $q^{pr}$ is in the form of the Chinese Restaurant Process prior. The sequential variational inference method for DP mixtures [16] is suitable for dealing with streaming data with overall complexity $O(NK)$, where $N$ is the number of data points and $K$ is the expected number of experts.

However, in the task-agnostic setting where the task assignment is evaluated at each time step, dynamics models are not independently sampled from a prior. Instead, the adjacent observations tend to belong to the same dynamics model, and there may exist temporal transition dependencies between different dynamics models. Therefore, we adopt the Markovian assumption when selecting the assignment prior $q_n^{pr}$ and propose a transition prior that conditions on the previous data point's

dynamics assignment $z_{n-1}$ as follows:

$$q_n^{pr} \propto \begin{cases} \sum_{i=1}^n \mathbf{1}\{z_{i-1} = z_{n-1}\}\rho_i(z_{ik}) + \mathbf{1}\{k = z_{n-1}\}\beta, & 0 \le k \le K_{n-1} - 1 \\ \alpha, & k = K_{n-1} \end{cases} \quad (3)$$

Here $\beta$ is the sticky parameter that increases the probability of self-transition to avoid rapidly switching between dynamics models [35]. Note that in Algorithm 1, we approximate the soft dynamics assignment $\rho_i(z_{ik})$ of the $i$th data pair with the hard assignment $z_i = \operatorname{argmax}_k \rho_i(z_{ik})$. The argmax/hard assignment approximation splits the streaming data and thus accelerates the stochastic update of model parameters and simplifies the form of transition prior. However, this involves a trade-off [36] between how well the data are balanced and the density accuracy.

For GPs, $\boldsymbol{\theta}$ is a set of kernel parameters $\{w_i, w_{i,1}, ..., w_{i,c}, \sigma_i\}_{i=1}^c$. Since it is hard to find a conjugate prior $G_0$ for $\boldsymbol{\theta}$, and we are considering a task-agnostic setting with limited prior knowledge, we use the likelihood $F((\tilde{\boldsymbol{x}}_n, \boldsymbol{y}_n)|\boldsymbol{\theta}_k)$ to approximate the integrals in (2), inspired by the auxiliary variable approach [37]. In order to create a GP expert for unobserved dynamics, we need to find the likelihood of a GP with no data [38]. Therefore, we initialize the kernel of the first GP dynamics model $M_0$ with a fixed reasonable point estimation $\boldsymbol{\theta}_{init}$ to evaluate the likelihood conditional on it. We then train a global GP dynamics model with all online collected data and use the updated parameters as the prior when initializing succeeding GP experts. To deal with the non-conjugate situation, we use stochastic batch optimization and update $\boldsymbol{\theta}$ with (4).

$$\boldsymbol{\theta}_k \leftarrow \boldsymbol{\theta}_k - \eta \sum_{i=0}^n \mathbf{1}\{z_i = k\}\nabla_{\boldsymbol{\theta}} \log p(\boldsymbol{y}_i | \tilde{\boldsymbol{x}}_i, \mathcal{D}, \boldsymbol{\theta}_k), \ \forall k = 0, ..., K_n - 1 \quad (4)$$

Note that the iterative updating procedure of dynamics assignment $z$ and dynamics-specific parameters $\boldsymbol{\theta}$ can be formulated as a variational expectation-maximization (EM) algorithm. In the variational E step, we calculate the soft assignment, which is the posterior probability conditioned on the dynamics parameters $\boldsymbol{\theta}$. In the M step, given each data pair's assignment, we update $\boldsymbol{\theta}$ for each dynamics model with the conjugate gradient method to maximize the log-likelihood.

## 4.2 Data Distillation with Inducing Points

GP dynamics models need to retain a dataset $\mathcal{D}$ for training and evaluation, for which the spatial and computational complexity increases as more data is accumulated. For instance, the prediction complexity in (1) is $O(N_k^3)$ due to the matrix inversion. To make the algorithm computationally tractable, we select a non-growing but large enough number of data points that contain maximum information to balance the tradeoff between algorithm complexity and model accuracy. In a model-based RL setting, the data distillation procedure is rather critical since the agent exploits the learned dynamics models frequently to make predictions in MPC, and thus the overall computational load heavily depends on the prediction complexity.

For each constructed dynamics model $k$, data distillation is triggered if the number of data points belonging to $M_k$ reaches a preset threshold $n_{distill}$. Define $\mathcal{D}_k = \{(z_i, \boldsymbol{x}_i, \boldsymbol{u}_i, \Delta\boldsymbol{x}_i) \mid z_i = k\}$. The $m$ inducing points $\mathcal{D}_{k,m}$ are selected while preserving the posterior distribution over $f$ as in exact GP in (1) by maximizing the following lower bound that is widely used in Sparse GP literature [17, 39]:

$$L = \sum_{i=1}^c \left[ \log \mathcal{N}(Y^i; 0, K_{nm}K_{mm}^{-1}K_{mn} + I\sigma_i^2) - \frac{1}{2\sigma_i^2} \operatorname{Tr}(K_{nn} - K_{nm}K_{mm}^{-1}K_{mn}) \right] \quad (5)$$

Note that the collected data depends on the learned dynamics model and the rolled-out policy, which introduces a nonstationary data distribution. In addition to the setting that the GP experts are trained online from scratch, treating inducing points as pseudo-points and optimizing (5) via continuous optimization may lead to an unstable performance in the initial period. Instead, we directly construct $\mathcal{D}_{k,m}$ by selecting real data points from $\mathcal{D}_k$ via Monte Carlo methods. Other criteria and optimization methods for selecting inducing points can be found in [40].

## 4.3 Expert Merge and Prune

Since each data point's dynamics assignment is only evaluated once, dynamics models may be incorrectly established, especially in early stages where the data numbers of existing dynamics

models are relatively small. Redundant dynamics models not only harm the prediction performance by blocking the forward and backward knowledge transfer but also increase the computational complexity when calculating $z$ (which requires iterating over the mixture). Therefore, we propose to merge redundant models and prune unstable ones with linear complexity w.r.t. the number of spawned dynamics models. Other methods for merging experts can be found in [41].

**Expert merging mechanism.** We say that a newly constructed dynamics model $M_K$ is in a burn-in stage if the number of data points belonging to it is less than a small preset threshold $n_{merge}$. At the end of the burn-in stage, we check the distances between $M_K$ with the older models. If the minimum distance obtained with $M_{k'}$ is less than the threshold $\epsilon$, we merge the new model $M_K$ by changing the assignments of data in $\mathcal{D}_K$ to $z = k'$. We use the KL-divergence between the predictive distributions evaluated at $\mathcal{D}_K$ conditioned on two dynamics models as the distance metric:

$$d(M_K, M_k) = \sum_{i \in \mathcal{D}_K} KL(p(f|\mathcal{D}_k, \tilde{\boldsymbol{x}}_i) \| p(f|\mathcal{D}_K, \tilde{\boldsymbol{x}}_i)) \tag{6}$$

**Expert pruning method.** We assume that in real-world situations, each type of dynamics lasts for a reasonable period. If a dynamics model $M_K$ accumulates data less than the merge trigger $n_{merge}$ when a new dynamics model is spawned, we treat $M_K$ as an unstable cluster and merge it with adjacent dynamics models based on the distance in (6).

# 5    Experiments: Task-Agnostic Online Model-Based RL

We present experiments in this section to investigate whether the proposed method (i) detects the change points of the dynamics given the streaming data, (ii) improves task detection performance by using the transition prior, (iii) automatically initializes new models for never-before-seen dynamics and identifies previously seen dynamics, and (iv) achieves equivalent or better task performance than baseline models while using only a fraction of their required data.

We use three non-stationary environments for our experiments, as shown in Figure 2. In each environment, the RL agent cyclically encounters several different types of dynamics. In `Cartpole-SwingUp`, we aim to swing the pole upright meanwhile keep the cart in the middle. The pole length $l$ and mass $m$ vary sequentially by alternating between four different combinations. `HalfCheetah` is a more complex control problem having larger state and control dimen-

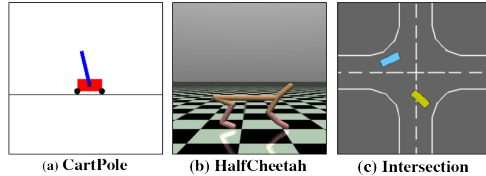

(a) CartPole    (b) HalfCheetah    (c) Intersection

Figure 2: Simulation environments

sions with the non-stationary dynamics induced by different torso loads. The goal is to make the half cheetah run as fast as possible. `Highway-Intersection` contains surrounding vehicles with different target destinations, and the ego vehicle needs to avoid collisions while heading to its goal. More detailed experiment settings are presented in Section S1 in the supplementary material.

We compare our method with five model-based baselines detailed as follows:

(a) **Single dynamics model**: We investigate the task performance of using a GP, a DNN, and an Attentative Neural Process (ANP) [42, 43] as the dynamics model. The GP is trained online from scratch, while the ANP and the DNN are first trained offline with data pre-collected in all dynamics that the agent may encounter online. All three models are updated online to account for the non-stationarity of environments after each batch of data is collected.

(b) **Mixture of DNNs**: We compare with another two pre-trained baselines using mixtures of DNNs as the dynamics model with DP prior (DPDNN) and the proposed transition prior (T-DNN), respectively. Considering that our method does not require an adaptation module for updating model parameters, we directly use the weights of a global DNN that is trained with all collected data as the prior instead of using a MAML-trained prior as in [8].

(c) **Meta-RL**: We use the MAML algorithm [20] to learn the dynamics model for model-based RL. After being pre-trained with several episodes, the meta-model is adapted online with recently collected data to deal with nonstationarity.

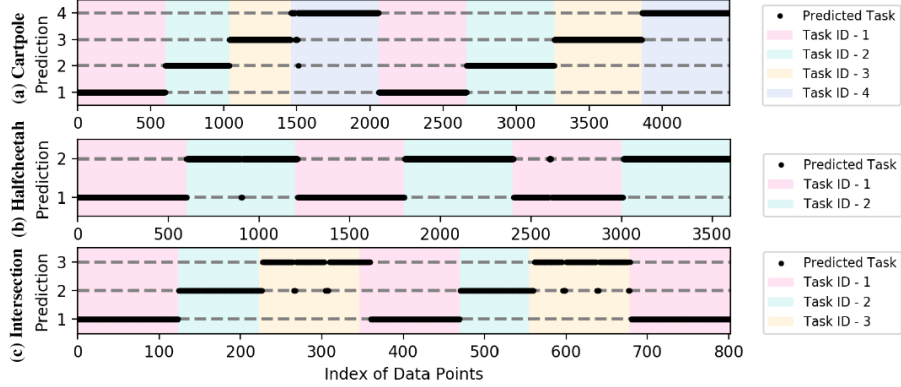

Figure 3: Dynamics assignments with the proposed transition prior. Our method successfully detects the dynamics shift. It allocates new components to model previously unseen types of dynamics and recalls stored models when encountering seen dynamics.

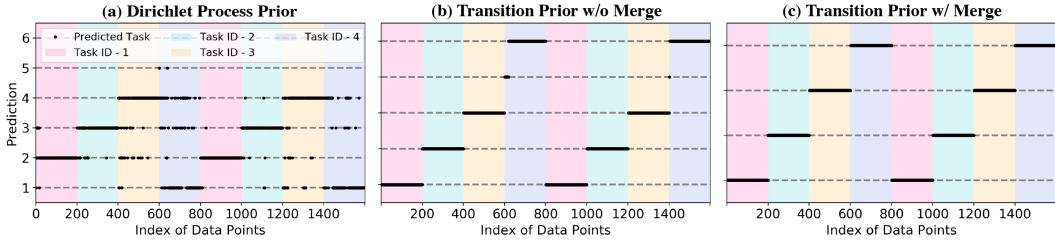

Figure 4: Ablation experiment results of the transition prior as well as the merge and prune mechanism in `Cartpole-SwingUp`. The proposed transition prior achieves more accurate dynamics assignments than the DP prior. The merge and prune mechanism successfully merges redundant dynamics models.

## 5.1 Task Switching Detection

In all three environments that contain sudden switches between different dynamics, our method detects the task distribution shift as visualized in Figure 3. In general, the predicted region of each type of dynamics matches that of the ground truth. We notice that there exists some delay when detecting change points and hypothesize that this may be due to the clustering property and sticky mechanism of the transition prior. Although DPs [44] have the so-called rich-gets-richer property, directly using a DP prior fails to capture the temporal relationship between data points and thus leads to inaccurate dynamics assignments (Figure 4 (a)). We also notice that wrong dynamics assignments result in unstable task performances with smaller rewards and larger variances, as shown in Section S3. There are some redundant dynamics models during online training (Figure 4 (b)) when not using the merge and prune mechanism. When comparing Figure 4 (b) and (c), we can see that our method successfully merges redundant dynamics models to the correct existing ones.

## 5.2 Task Performance

We evaluate the task performance in terms of the accumulated rewards for each type of dynamics, as displayed in Figure 5. Since a separate model is initialized and trained from scratch whenever a new type of dynamics is detected, our method's reward oscillates during the initial period after the dynamics shift (especially in Figure 5 (a)). However, our method performs well in a new type of dynamics after collecting just a handful of data points that are far less than DNN-based methods. For example, for each type of dynamics in `Cartpole-SwingUp`, our method converges after collecting 600 data points, while DNN may need around 1500 data points [12]. By learning the latent dynamics assignment, our method quickly adapts when it encounters previously seen dynamics by shifting to the corresponding dynamics model. These previously learned models are further improved by being updated with the newly collected data according to their dynamics assignments. Additionally, our method is more robust than all the baselines and has a smaller variance at convergence.

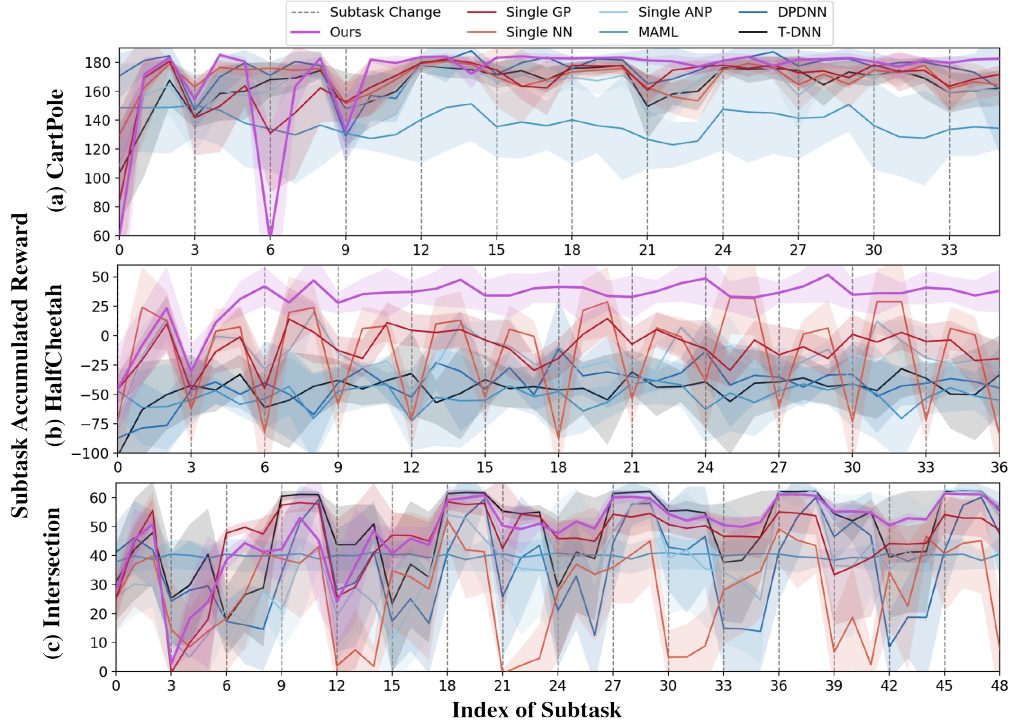

Figure 5: Accumulated rewards in three nonstationary environments. A subtask is analogous to an episode. Each vertical dotted line indicates a dynamics switch. Our method is more robust, data-efficient, and has higher rewards.

Using a single dynamics model automatically preserves global dynamics properties. Therefore using a single GP without pre-training does not suffer apparent performance degradation when the dynamics switch. Since both the DNN and the ANP use pre-trained models, they achieve equivalent (in complex `Highway-Intersection`) or better (in simple `Cartpole-SwingUp`) performance as our method when encountering new dynamics. However, the three single dynamics model baselines fail to consistently succeed and oscillate a lot as the environment getting more complex. For example, when using a DNN in the unprotected left-turn scenario in `Highway-Intersection`, the RL agent keeps colliding with the other vehicle. Although the ANP is a DNN approximation of the GP and can deal with multiple tasks itself [45], its performance still oscillates, and thus it cannot robustly handle the multimodal task distribution. The online adaptation mechanism causes the single model to tend to overfit to the current dynamics. The overfitting problem further harms the quick generalization performance since the adjacent dynamics are substantially different, and previously adapted model parameters may be far from the optimal ones for the new dynamics. Additionally, without task recognition, the model suffers from the forgetting problem due to not retaining data and parameters of previously seen dynamics.

Figure 5 also show that our method outperforms the three baselines that aim to handle nonstationarity. The model-based RL with MAML is easily trapped in local minima determined by the meta-prior. For instance, the RL agent in `Highway-Intersection` learns to turn in the right direction but fails to drive to the right lane. In our experiments, MAML also cannot quickly adapt to substantially different dynamics. This is because MAML suffers when a unimodal task distribution is not sufficient to represent encountered tasks. The model-based RL with a DP mixture of DNNs performs slightly better than a single DNN but still oscillates due to inaccurate task assignments, as in Section S3. T-DNN performs similarly to DPDNN and still underperforms our method due to the inaccuracy of DNN model predictions with limited data.

# 6 Discussion

We propose a scalable online model-based RL method with an infinite mixture of GPs to handle real-world task-agnostic situations with limited prior knowledge. Our method achieves quick adaptation and avoids pre-training by using data-efficient GPs as dynamics models and avoids catastrophic forgetting by retaining a mixture of experts. Our model performs well when dynamics are substantially different by constructing a multimodal predictive distribution and blocking harmful knowledge transfer via task recognition. We propose a transition prior to explicitly model the temporal dependency and thus release the assumption that tasks are independent and identically distributed. Additionally, our method detects the dynamics shift at each time step, so it is suitable for situations with unknown task delineations. We learn the mixture via online sequential variational inference that is scalable to extensive streaming data with data distillation and the merge and prune technique. Since computing the posterior of a GP becomes intractable as the data size and data dimension increase, replacing GPs with Neural Processes [42, 46] would be an interesting direction to explore. Another direction would be to incorporate meta-learning into our method to better leverage the commonly shared information across the different dynamics.

## Broader Impact

This work is a step toward General Artificial Intelligence by eliminating the pre-train stage and equipping reinforcement learning agents with the quick-adaptation ability. The proposed method could be applied in a wide range of applications when the prior knowledge is not accessible or not beneficial, including space rover navigation, rescue robot exploration, autonomous vehicle decision making, and human-robot interaction.

Our research increases algorithmic interpretability and transparency for decision-making by providing inferred task assignments. It also enhances the algorithm's robustness by explicitly separating different types of tasks and thus increases users' trust. Additionally, our research improves algorithmic fairness by not relying on prior knowledge that may contain human-induced or data-induced biases. At the same time, our research may have negative impacts if misused. The potential risks are listed as follows: (1) Over-trust in the results (e.g., the task assignments) may lead to undesirable outcomes. (2) If used for illegal purposes, the model may instead enlarge the possible negative outcome due to its explainability. (3) The task assignment results may be misinterpreted by those who do not have enough related background. (4) The evolving online nature of the method may increase the uncertainty and thus mistrust of human collaborators. Note that our method inherits the possible positive and negative impacts of reinforcement learning, not emphasized here.

To mitigate the possible risks mentioned above, we encourage research to (1) investigate and modulate the negative impacts of the inaccurate information provided by algorithms, (2) understand how humans interact with evolving intelligent agents in terms of trust, productivity, comfort level, (3) find out the impact of using our model in real-world tasks.

## Acknowledgments and Disclosure of Funding

Thanks go to Mr. Keegan Harris for revision and advice. The authors gratefully acknowledge the support from the Manufacturing Futures Initiative at Carnegie Mellon University, made possible by the Richard King Mellon Foundation. The ideas presented are solely those of the authors.

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
