[Supplementary Material · TA-mbrl-gpmm-camera-ready-suppl.pdf]

# Supplementary Material

## S1 Simulation Envirionments

We present details of the three non-stationary simulation environments in this section. Since we deal with real-world physical tasks, each kind of task has its own dynamics type. The RL agent encounters different types of dynamics sequentially in our setting. The switch of dynamics is assumed to finish within a single timestep. Each type of dynamics may last for several episodes, and we call each episode as one subtask since the dynamics type within an episode is invariant. Note that our method is not restricted by the episodic assumption since the task index and task boundary are unknown. A summary of the key parameters is shown in Table S1.

### S1.1 Changeable Pole Length and Mass of CartPole-SwingUp

`CartPole-SwingUp` consists of a cart moving horizontally and a pole with one end attached at the center of the cart. We modify the simulation environment based on the OpenAI Gym environment [1]. Different types of dynamics have different pole mass $m$ and pole length $l$. In our setting, the agent encounters four types of dynamics sequentially with the combinations $(m, l)$ as $(0.4, 0.5)$, $(0.4, 0.7)$, $(0.8, 0.5)$, $(0.8, 0.7)$. Denote the position of the cart as $x$ and the angle of the pole as $\theta$. The state of the environment is $\boldsymbol{x} = (x, \dot{x}, \cos\theta, \sin\theta, \dot{\theta})$ and action $\boldsymbol{u}$ is the horizontal force applied on the cart. Therefore, the GP input is a 6 dimensional vector $\tilde{\boldsymbol{x}} = (\boldsymbol{x}, \boldsymbol{u})$. The GP target is a 5 dimensional vector as the state increment $\boldsymbol{y} = \Delta\boldsymbol{x} = (\Delta x, \Delta \dot{x}, \Delta\cos\theta, \Delta\sin\theta, \Delta\dot{\theta})$.

### S1.2 Varying Torso Mass in HalfCheetah

In `HalfCheetah`, we aim to control a halfheetah in flat ground and make it run as far as possible. The environment is modified based on MuJoCo [2]. We notice that the torso mass $m$ significantly affects the running performance. Therefore, we create two different types of dynamics by changing the torso mass $m$ iteratively between $14$ and $34$ to simulate real-world delivery situations. We denote the nine joints of a halfcheetah as (root_x, root_y, root_z, back_thigh, back_shin, back_foot, front_thigh, front_shin, front_foot). The state $\boldsymbol{x}$ is 18-dimensional consisting of each joint's position and velocity. The action $\boldsymbol{u}$ is 6-dimensional, including the actuator actions applied to the last six physical joints. Therefore, the GP input $\tilde{\boldsymbol{x}} = (\boldsymbol{x}, \boldsymbol{u})$ is a 24-dimensional, and the GP target is the 18-dimensional state increment $y = \Delta\boldsymbol{x}$.

### S1.3 Dynamic Surrounding Vehicles in Highway-Intersection

The `Highway-Intersection` environment is modified based on highway-env [3]. We adapt the group modeling [4] concept and treat all the other surrounding vehicles' behaviors as part of the environment dynamics. In addition to modeling the interactions (analogous to the effect of ego vehicle action to other vehicles), the mixture model also needs to learn the ego vehicles' dynamics (analogous to the action's effect on ego vehicle). For simplicity, we only experiment with environments containing only one surrounding vehicle. Note that the multi-vehicle environment can be generated by following the same setting but requires modification of pipelines. For example, to evaluate the posterior probability of all surrounding vehicles from GP components, we can query the mixture model multiple times and then calculate the predictive distribution for each vehicle.

We consider an intersection with a two-lane road. The downside, left, upside, and right entry is denoted with index 0,1,2,3, respectively. The ego vehicle $A_0$ has a fixed initial state as the right lane of entry 0 and a fixed destination as the right lane of entry 1. In other words, $A_0$ tries to do a left turn with high velocity and avoid collision with others. Each type of dynamics has different start and goal positions of the surrounding vehicle $A_1$. $A_0$ encounters three types of interactions with the combinations of $A_1$'s start entry and goal entry as $(2, 1)$, $(2, 0)$ and $(1, 2)$. Note that when $A_1$ emerges at entry 2 and heading to entry 0, $A_0$ faces a typical unprotected left turn scenario.

Denote the positions and heading of a vehicle as $(x, y, h)$. The state of $A_0$ is $(x, y, \dot{x}, \dot{y}, \cos h, \sin h)$ in the word-fixed frame. The state of $A_1$ is $(x_{rel}, y_{rel}, \dot{x}_{rel}, \dot{y}_{rel}, \cos h_{rel}, \sin h_{rel})$ evaluated in the body frame fixed at $A_0$. We directly control the ego vehicle $A_0$'s acceleration $a$ and steering angle $\theta$.

Table S1: Simulation Environment Details. Each type of dynamics last for 3 episodes.

| Environment | CartPole | HalfCheetah | Intersection |
|---|---|---|---|
| State Dimension | 5 | 18 | 12 |
| Action Dimension | 1 | 6 | 2 |
| Episode Length | 200 | 200 | 40 |
| Simulation Interval (s) | 0.04 | 0.01 | 0.1 |
| Early Stop | True when $x$ out of limit | False | True when Collision |
| No. episodes / Dynamics | 3 | 3 | 3 |

Table S2: Model Parameters.

| Parameter | CartPole | HalfCheetah | Intersection |
|---|---|---|---|
| concentration parameter $\alpha$ | 0.1 | 1.5 | 0.5 |
| sticky paramter $\beta$ | 1 | 1 | 1 |
| initial noise $\sigma_i, i = 1, ...c$ | 0.001 | 0.1 | 0.001 |
| initial output scale $w_i, i = 1, ...c$ | 0.5 | 10.0 | 0.5 |
| initial lengthscale $1/w_{i,j}, i, j = 1, ...c$ | 1.0 | 1.0 | 1.0 |
| merge KL threshold $\epsilon$ | 20 | 10 | 70 |
| merge trigger $n_{merge}$ | 15 | 5 | 10 |
| data distillation trigger $n_{distill}$ | 1500 | 2000 | 1500 |
| inducing point number $m$ | 1300 | 1800 | 1300 |
| GP update Steps / timestep | 10 | 5 | 10 |
| learning rate | 0.1 | 0.1 | 0.1 |
| discount $\gamma$ | 1 | 1 | 1 |
| MPC plan horizon | 20 | 15 | 20 |
| CEM popsize | 200 | 200 | 200 |
| CEM No. elites | 20 | 10 | 20 |
| CEM iterations | 5 | 5 | 5 |

Therefore, the GP input $\tilde{x}$ is a 14 dimensional vector consisting of the $A_0$'s state and action as well as $A_1$'s state. The GP target is the increment of the ego vehicle's and the other vehicle's states.

## S2 Method Details

The computing infrastructure is a desktop with twelve 64-bit CPU (model: Intel(R) Core(TM) i7-8700K CPU @ 3.70GHz) and a GPU (model: NVIDIA GeForce RTX 2080 Ti). Since our method is pre-train free, we do not need to collect data from different dynamics beforehand. However, for most of the baselines, we collect data to pre-train them as detailed in Section S2.2. In our online setting, there is no clear boundary between training and testing. More concretely, at each time step, our model is updated by the streaming collected data and evaluated in MPC to select the optimal action.

### S2.1 Parameters of Our Method

We list key parameters of our proposed method in Table S2. The concentration parameter $\alpha$ controls the generation of new GP components. The larger the $\alpha$, the more likely a new GP component is spawned. The sticky parameter $\beta$ increases the self-transition probability of each component. The initial parameter $\boldsymbol{\theta}_0$ for a new GP consists of initial noise $\sigma_i$, initial output scale $w_i$ and initial lengthscale $1/w_{i,j}$. Larger lengthscale and output scale help alleviate the overfitting problem by increasing the effect from the away points. HalfCheetah has a higher state dimension than the other two and thus has larger $n_{distill}$ and $m$. The planning horizon of Intersection is half of the total episode length since the MPC needs to do predictive collision check to achieve safe navigation. For the hyperparameter selection, we randomly search in a coarse range first and then do a grid search in a smaller hyperparameter space.

Table S3: Parameters of the DNN, ANP and MAML Baselines.

| Baseline | Parameter | CartPole | HalfCheetah | Intersection |
|---|---|---|---|---|
| | pre-train episodes / dynamics | 10 | 10 | 10 |
| DNN | gradient steps | 200 | 100 | 100 |
| | optimizer | Adam | Adam | Adam |
| | learning rate | 0.0005 | 0.001 | 0.008 |
| | hidden layers | 2 | 2 | 2 |
| | units per layer | 256 | 500 | 256 |
| | minibatch size | 512 | 256 | 128 |
| ANP | gradient steps | 200 | 800 | 100 |
| | optimizer | Adam | Adam | Adam |
| | learning rate | 0.0005 | 0.001 | 0.0005 |
| | hidden layers | [256, 128, 64] | [512, 256, 128] | [256, 128, 64] |
| | minibatch size | 1024 | 64 | 1024 |
| | context number | 100 | 100 | 100 |
| | target number | 25 | 25 | 25 |
| | latent dimension | 64 | 128 | 64 |
| MAML | hidden layers | 2 | 2 | 2 |
| | units per layer | 500 | 500 | 500 |
| | step size $\alpha$ | 0.01 | 0.01 | 0.01 |
| | step size $\beta$ | 0.001 | 0.001 | 0.001 |
| | meta steps | 200 | 100 | 100 |
| | adapt learning rate | 0.001 | 0.001 | 0.001 |
| | adapt step | 10 | 10 | 10 |
| | meta batch size | 1 | 1 | 1 |

## S2.2 Parameters of Baselines

The critical parameters of using a DNN, an ANP [5, 6], and a MAML [7] are shown in Table S3. The parameters for a single GP baseline and the concentration parameter $\alpha$ of the DPNN baseline are the same as the corresponding ones of our proposed method, as in Table S2. The DNN parameters in DPNN baseline is the same as the parameters of a single DNN, as in Table S3. Note that except for the baseline using a single GP as dynamics models, all the other baselines require to collect data to pre-train the model. In our setting, we collect ten episodes from each type of dynamics as the pre-train dataset. The parameters for baselines are all carefully selected to achieve decent and equitable performance in our nonstationary setting. For instance, since HalfCheetah has a larger state dimension than the other two, it has larger units per layer in DNN and latent dimension in ANP.

To adapt the MAML method, in addition to the pre-train free assumption, we further release the assumption that the task boundaries between different dynamics are unknown during the pre-train procedure. Note that MAML is pre-trained with the same amount of data as the other baselines to guarantee a fair comparison. The performance of MAML may increase if collecting more data. During the online testing period, the adapt model copies meta-model's weights and updates its weights with recently collected data at each timestep.

## S3 Additional Experiment Results

### S3.1 Dynamics Assignments with Dirichlet Process Prior

We show that using pure DP prior is not sufficient to capture the dynamics assignments of streaming data in real-world physical tasks by visualizing the cluster results in CartPole-SwingUp, as in Figure 4. In this section, we show more statistics about the dynamics assignments with DP prior by comparing the performance of DPNN and our method in Figure S1.

In CartPole-Sqingup, we can see that our method can accurately detect the dynamics shift and cluster the streaming data to the right type of dynamics. However, when using DPNN, the more types of dynamics encountered, the less accurate the assignments are. In Highway-Intersection, our method sometimes cluster the data points into the wrong dynamics. We hypothesize that this

(a) `CartPole-Sqingup`  (b) `Highway-Intersection`

Figure S1: Correct Assignment Percentage using DPNN and our method. Our method ourperforms DPNN in terms of task assignments.

Table S4: Reward Mean and Standard Deviation (std). The bold numbers indicate the maximum means and minimum stds in each task.

| | | CartPole | | | | HalfCheetah | | Intersection | | |
|---|---|---|---|---|---|---|---|---|---|---|
| | | Task1 | Task2 | Task3 | Task4 | Task1 | Task2 | Task1 | Task2 | Task3 |
| Our | mean | 177.62 | **183.10** | **181.37** | **182.54** | **36.11** | **34.14** | 60.66 | **54.77** | **52.48** |
| Method | std | 6.61 | **3.29** | **1.86** | **1.10** | **9.22** | 13.05 | 1.29 | **1.80** | **0.77** |
| GP | mean | 178.04 | 169.46 | 174.45 | 171.67 | 2.70 | -21.26 | 53.73 | 39.23 | 44.25 |
| | std | **3.40** | 7.39 | 8.94 | 6.12 | 14.70 | 24.49 | 6.56 | 20.31 | 6.91 |
| DNN | mean | 176.57 | 164.72 | 177.74 | 164.21 | 28.89 | 0.53 | 43.42 | 2.31 | 46.62 |
| | std | 8.14 | 16.41 | 5.21 | 7.24 | 35.02 | 15.60 | 9.73 | 7.34 | 6.12 |
| ANP | mean | 175.33 | 170.74 | 171.29 | 160.87 | -4.27 | -20.70 | **62.68** | 44.04 | 36.91 |
| | std | 6.61 | 4.44 | 6.68 | 8.34 | 16.80 | 22.09 | **0.75** | 17.95 | 19.86 |
| MAML | mean | 144.91 | 150.80 | 127.55 | 134.36 | -70.49 | -51.32 | 39.29 | 39.14 | 40.31 |
| | std | 28.22 | 18.74 | 32.82 | 13.95 | 36.57 | 12.13 | 3.36 | 4.21 | 5.89 |
| DPNN | mean | **187.24** | 182.52 | 180.20 | 161.58 | -42.88 | -39.30 | 59.97 | 46.89 | 18.64 |
| | std | 5.03 | 6.82 | 3.09 | 45.27 | 16.37 | **8.14** | 2.92 | 15.33 | 22.42 |

may be due to the overlap in the spatial domain of different interactions. However, our method still outperforms the DPNN in terms of dynamics assignments. DPNN can only stably identify the second task, and either frequently generate new clusters for the first and third task or identify them as the second task. We notice that the clustering accuracy of DPNN heavily relies on the number of previous states concatenated (the length of the short term memory) [8]. To make the dynamics assignment of DPNN more stable, in our setting, we use 50 (1/4 episode length) previous data points to determine the dynamics assignment in `CartPole` and 20 (1/2 episode length) in `Intersection`.

## S3.2 Will Dynamics Assignments Affect Task Performance?

To investigate whether the correct dynamics assignments improve the task performance, we compare the accumulated subtask rewards of our method, the single GP baseline, and the DPNN baseline. The single GP baseline is the ablation version of our method without dynamics assignments. In other words, using a single GP indicates clustering different dynamics into a single cluster. The DPNN has less accurate dynamics assignments than our method, as detailed in Section S3.1.

Table S4 and Figure 5 show that our method has higher rewards and smaller variances than the baselines in most situations. Since our method performs better than the single GP baseline in all three nonstationary environments, it shows that the dynamics recognition help increase the task performance. Note that DPNN has higher rewards than our method in the first task of `CartPole-SwingUp`. We hypothesize that this may be due to the pre-train procedure of DPNN. However, our method outperforms DPNN in all the other dynamics, which indicates that accurate dynamics assignments help improve task performances.

## S4  Model Derivation

In this part, we highlight the derivation of the soft assignment $\rho_n(z_{nk})$ in 2 that was originally derived in [9] Eq.12-16 to ease reading. Note that we highly recommend reading [9] for thorough understanding. The posterior distribution $p_n(z_{0:n}, \boldsymbol{\theta}|\mathcal{D})$ for the assignments $z_{0:n}$ and model parameter $\boldsymbol{\theta}$ can be written as product of factors

$$p_n(z_{0:n}, \boldsymbol{\theta}|\mathcal{D}) \propto p((\tilde{\boldsymbol{x}}_n, \boldsymbol{y}_n)|\boldsymbol{\theta}_{z_n})p(z_n|z_{0:n-1})p_{n-1}(z_{0:n-1}, \boldsymbol{\theta}|\mathcal{D}_{0:n-1})$$

$$\propto p(\boldsymbol{\theta})\prod_{i=i}^{n} p((\tilde{\boldsymbol{x}}_i, \boldsymbol{y}_i)|\boldsymbol{\theta}_{z_i})p(z_i|z_{1:i-1}) \tag{S1}$$

Since S1 is in factorized form, we can apply Assumed Density Filtering (ADF) to approximate the posterior of the first 0 to n data pair with $\hat{q}_n(z_{0:n}, \boldsymbol{\theta}) = \prod_{k=0}^{\infty} \gamma_n(\boldsymbol{\theta}_k) \prod_{i=0}^{n} \rho_n(z_i)$. The approximate posterior for the $n+1$-th data pair is thus

$$\hat{p}_{n+1}(z_{0:n+1}, \boldsymbol{\theta}|\mathcal{D}_{0:n+1}) \propto p((\tilde{\boldsymbol{x}}_{n+1}, \boldsymbol{y}_{n+1})|\boldsymbol{\theta})p(z_{n+1}|z_{0:n})\hat{q}_n(z_{0:n}, \boldsymbol{\theta}) \tag{S2}$$

$$\hat{q}_{n+1}(z_{0:n+1}, \boldsymbol{\theta}) = \underset{q_{n+1}\in\mathcal{Q}_{n+1}}{\arg\min} \ KL\big(\hat{p}_{n+1}(z_{0:n+1}, \boldsymbol{\theta}|\mathcal{D}_{0:n+1})\|q_{n+1}(z_{0:n+1}, \boldsymbol{\theta})\big) \tag{S3}$$

With the mean field assumption, the optimal distribution for the new observation is given in 2.