[Reviews · NeurIPS 2020]

Review 1

Summary and Contributions: *** Update: I have read the author response, which was very nice. I maintain a high opinion of this paper, and think it should be accepted. *** This paper proposes a Bayesian model to adapt to nonstationary dynamics in RL problems. The paper draws on DPs, GPs, an other ideas to create a principled model that is robust and data efficient. A small but well-done experimental section compares against several natural baselines, and demonstrates the superiority of the approach.

Strengths: I liked this paper. This model feels principled, clearly defined, and technically sound. I felt that the paper was well-written, with many design choices that seemed solid. The posterior inference algorithm is complex, but not unnecessarily so. The experiments were small, but reasonably well done. I especially appreciated the inclusion of several natural baselines. The method seems to work well, and does the job it was designed to do.

Weaknesses: The model is fairly complex, and some of the design choices seem a bit ad hoc. As this is a Bayesian model, there ought to be principled posterior distributions for every quantity; your "algorithm" should then be inference in those posterior distributions. For example, you have somewhat hack-ish methods of splitting and merging experts, but this could be done using appropriate MCMC proposals, such as birth and death kernels. Likewise, the argmax assignments feel like an approximation to a true posterior. In general, I think the paper could be strengthened by clearly laying out the normative quantities, and then discussing how your algorithmic choices approximate those, and what the tradeoffs of such approximations are likely to be.

Correctness: Yes

Clarity: Yes - I found this to be a very well-written paper.

Relation to Prior Work: Yes

Reproducibility: Yes

Additional Feedback: Well done!


Review 2

Summary and Contributions: The paper proposes a continual online model-based RL approach that does not require pre-training to solve task-agnostic problems with unknown task boundaries. In order to achieve this, the authors mainly maintain an infinite mixture of experts and represent each different type of dynamics with a GP (Gaussian process). With the dynamic models, it uses MPC to make decisions. And to adapt the changes of dynamics, it uses a transition prior and proposes the expert merge and prune mechanism.

Strengths: The framework proposed by the authors can handle online nonstationary tasks without requiring task delineation or depending on a pre-trained, which is not able to be realized in meta-learning framework. Also, it is closer to the practical scenarios in application. Since it utilizes GP rather than DNN as the dynamic model, it is more explicable, and the time complexity is estimable. According to the experiments shown in the paper, the framework outperforms other algorithms like MAML in Cartpole, HalfCheetah and Intersection in general. Nowadays, DNN is widely used in different fields and in various tasks, I think this work will give us some inspiration and look back to some classic methods.

Weaknesses: I think more detailed derivation process should be included in the paper. Additionally, the environments tested in the paper seems not that complex, I mean, it will be better if experiments in environments with higher dimensions can be performed. As it’s declared in the paper, the performance of model-based methods heavily depends on the accuracy of the learned dynamic models. So in higher dimension environments, does the infinite mixture of GP still outperform the previous dynamic models like DNN?

Correctness: As far as I concerned, I don’t think there exists claims or methods incorrect in the paper. Also, the empirical methodology is reasonable.

Clarity: From my point of view, the paper is well written. The authors firstly introduce the background knowledge of continual learning and model-based RL and point out the problems remaining in the field of learning in online nonstationary tasks. And then the authors represent their framework from infinite mixture of GP to expert merge and prune mechanism. Finally, it compares their framework with five model-based baselines in experiment part with understandable charts and shows the strength of their algorithm. But it’s noted that the derivation process of some equations like equation 2, in my opinion, should at least be included in the supplemental material rather than just mentioned in the references.

Relation to Prior Work: The authors mainly compare their work with prior work in the field of meta-learning and continual learning and clearly explain the difference of this work with the previous work.

Reproducibility: Yes

Additional Feedback: As mentioned above, the paper will be better if more complex environment and more detailed derivation processes can be included.


Review 3

Summary and Contributions: This paper proposes a method for continuous learning in reinforcement learning where tasks regularly change, tasks remain available for some time and can be recurring but changepoints are not provided to the algorithm. The proposed approach is model based and the model of the dynamics is an infinite mixture of Gaussian processes. The model is compared to several baselines on Cartpole-SwingUp, HalfCheetah and Highway- Intersection where tasks are created by changing the dynamics of the environment. Experiments evaluate change point detection and new/previous task identification, task performance and data efficiency with respect to baselines.

Strengths: The paper is relatively easy to read. Experiments are well described, show the advantages of the proposed model, baselines are well chosen and ablation studies are also performed. I like that the approach is not based on deep learning, it is good to have some variety. The problem addressed is important.

Weaknesses: The impact of hyperparameters or good values are not discussed, unless I missed it. The evolution of the number of models in the mixture is not discussed. See Additional feedback. The experiments are performed on simple environments, so it is not clear whether the model works in complex environments. The idea is original as far as I can tell but the paper relies on several previous works for steps of the learning algorithm which sometimes make it hard to understand the choices made.

Correctness: The paper appears to be correct although I wonder whether there is a typo in equation 3. I was also not convinced of the superior data efficiency of the proposed approach. In my opinion this is not well established by the experiments, although I want to stress that superiority was established on other criteria and I do not think this is a major issue. More details below. In equation 3, beta is described as a sticky parameter that increases the probability of self-transition. However, it seems to be added to all sums regardless of the value of successive states z. Could there be a typo in the equation, for example a missing rho_{n-1}(z_ik), or am I missing something? In the experiments, I would suggest comparing the proposed method to DP mixture of DNNs with the transition prior, to better establish whether the GP or the transition (or both) are responsible for the improved performance. With the current experiments, I think it is not possible to claim that the proposed method is more data efficient than [8].

Clarity: The paper is overal clear and well written. I have a few suggestions to make it even easier to understand and/or fix some minor inconsistency. There is no need for the authors to answer to these points as I think the paper is already rather clear. I am unsure what Figure 1 represents. I might have missed it, but I think pi is not defined. Furthermore, for me, the graphical model representation seems to match the flow of the online learning algorithm rather than the generative distribution. To give some examples, in Figure 1a, model g_t depends on g_{t-1} and x_{t-1}. Furthermore, the notation in figure 1b might not be coherent with the text of section 2: I suspect that the label of the x-axis should be x_tilde, so (x,u) rather than x. In the definition q^{pr} in line 168, it was not clear to me what the subscript of the sum mean. Is it any variable in the range? Is it any subsequence? Any possible realization of these variables (what I assumed) ? In equation (4), I understand that z_t is the value assigned previously in the algorithm. I would suggest to use a different notation for this value than for the variable in equation (2,3) which represents the variable.

Relation to Prior Work: I was not familiar with all related works. In my opinion relationships to previous work and differences are well described. My only concern here is that [8,9] are said not to work in practice based on a publication [23] that was 10 years before [8,9]. That being said [8] is a baseline in the experiment so it is not a big concern.

Reproducibility: Yes

Additional Feedback: Models will never be re-evaluated for merge after being seen a sufficient amount of time. Couldn't this lead to an explosion in the number of models? Is there any theoretical guarantee on the rate of increase of the number of models? possible typo: l 94: from --> form ---------------------- Dear authors, thank you very muck for the detailed feedback. In particular I am happy to see the additional experiment I suggested. I have updated my score to a 7, as I feel the paper is worth accepting with the modifications you propose. It might also be interesting to mention mixture compression and the points discussed with R2 as potential improvements.

[Author Response · NeurIPS 2020]

We thank the reviewers for their thorough reviews and valuable feedback. We will address the concerns as follows.

**(R2): Approximate Inference and trade-offs.** We approximate the posterior of assignments and model parameters
via MAP to fulfill the time constraint in online learning tasks. The argmax/hard assignment approximation splits the
streaming data and thus accelerates the stochastic update of model parameters and simplifies the form of transition prior.
However, this involves a trade-off between how well the data are balanced and the density accuracy. The proposed
split-and-merge mechanism explicitly leverages the expressiveness of GPs while lacking theoretical guarantees, e.g.,
escaping local modes. We will clarify in revision. We appreciate the suggestions and will explore split-merge MCMC
[Jain and Neal, 2004] and memorized online VI [Hughes and Sudderth, 2013] in the future.

**(R3, R4): More complex environments.** The exact equivalence between infinitely wide DNNs and GPs was derived
by Lee et al. [2017]. Considering that DNNs require optimizing a large number of parameters, in general, the data
efficiency of GPs should be higher than DNNs even in high-dimensional environments. However, the space and
computational complexities of GPs dramatically increase along with the input dimension, which may deteriorate
real-world performance in complex environments. We plan to incorporate advanced models, including Deep Kernel
Learning and Neural Processes, into our method to efficiently handle high dimensional data in future work.

**(R3): Lack of derivation of equation 2. (R4): Sometimes hard to understand algorithm design choices.** To make
our algorithm easier to follow, we will (i) add the derivation of equation 2 and show that the optimal $\rho_n(z_{nk})$ is
the marginal distribution that minimizes $KL(p_n(z_{0:n}, \boldsymbol{\theta}|\mathcal{D})||\hat{q}_n(z_{0:n}, \boldsymbol{\theta}))$, where $p_n$ is the sequentially decomposed
posterior and $\hat{q}_n$ is a sequence of variational approximations (line 123); (ii) extend algorithm descriptions and clarify
correspondences between motivations and algorithm design choices in Section 4. For instance, as mentioned by **R2**, we
will clarify that we use $z_i$ to approximate $\rho_i$ to simplify the transition prior's form and decrease the computation burden.

**(R4): Impact of hyperparameters or good values.** We summarized the key parameters and the best selections in
Table S2 and described the impact of DP and GP parameters in Section S2.1 line S57-S65. For the hyperparameter
selection, we randomly search in a coarse range first and then do a grid search in a smaller hyperparameter space. We
will clarify the good value selection and describe the impact of MPC parameters in revision.

**(R4): Evolution of the mixture size.** Since the mixture size increases when a new type of task is detected, the
increasing rate depends on the natures of real-world applications, e.g., environment changing pace. We agree that the
mixture size may explode, which is a side effect of the desired capability of modeling an infinite number of tasks. The
proposed merge and prune method helps control the increasing rate by eliminating redundant ones. We will add a
size modulation mechanism in revision, such as setting a threshold based on computational burden and periodically
compressing the mixture based on distance measures. We will focus on mixture compression methods in future work.

**(R4): A typo in equation 3.** We are grateful that you pointed out this typo. We will update the first case in equation 2
to $q_n^{pr} \propto \sum_{i=1}^{n} \mathbf{1}\{z_{i-1} = z_{n-1}\}\rho_i(z_{ik}) + \mathbf{1}\{k = z_{n-1}\}\beta$, if $0 \le k \le K_{n-1} - 1$, where $\beta$ is only added to the prior
of model $k = z_{n-1}$. We confirm that the algorithm was implemented correctly by checking the code, and thus the
experiment results still hold. The subscript of summation in $q_n^{pr}$ refers to the index of collected data points. We will fix
the notations in Figure 1 with $\pi \to q^{pr}$ and $x \to \tilde{x}$.

**(R4): Data efficiency.** The DPDNN baseline is different from [8] in terms of DNN initialization (line 220-223). Our
*pre-train free* method achieves higher rewards than DPDNN pre-trained in each task (line S76-S81), which shows that
our method is more data-efficient than DPDNN. As suggested, we add a DNN mixture with the transition prior baseline
(T-DNN) that is also pre-trained. The results show that T-DNN underperforms our method due to the inaccuracy of
DNN model predictions with limited data. This ablation study is valuable and will be discussed in the paper.

**(R4): Related work.** Reference [23] supports that model-free methods may be impractical due to data inefficiency in
real applications. We will clarify this and emphasize that [8] is a model-based meta-learning method in Section 2.

# References

Sonia Jain and Radford M Neal. A split-merge markov chain monte carlo procedure for the dirichlet process mixture
model. Journal of computational and Graphical Statistics, 13(1):158–182, 2004.
Michael C Hughes and Erik Sudderth. Memoized online variational inference for dirichlet process mixture models. In
Advances in Neural Information Processing Systems, pages 1133–1141, 2013.
Jaehoon Lee, Yasaman Bahri, Roman Novak, Samuel S Schoenholz, Jeffrey Pennington, and Jascha Sohl-Dickstein.
Deep neural networks as gaussian processes. arXiv preprint arXiv:1711.00165, 2017.


[Meta-Review · NeurIPS 2020]

Reviewers agreed the paper contains interesting and sound contributions to an important problem, and is generally well written, although the model is fairly complex and the experimental domains are a bit simple. The authors are encouraged to provide further details to justify/explain certain algorithmic choices, include some of the key derivation steps (maybe with details in the appendix), and augment the experiments (like those in the rebuttal).